# Impact of Glass Irradiation on Laser-Induced Breakdown Spectroscopy Data Analysis

**DOI:** 10.3390/s23020691

**Published:** 2023-01-07

**Authors:** Londrea J. Garrett, Bryan W. Morgan, Miloš Burger, Yunu Lee, Hyeongbin Kim, Piyush Sabharwall, Sungyeol Choi, Igor Jovanovic

**Affiliations:** 1Department of Nuclear Engineering and Radiological Sciences, University of Michigan, Ann Arbor, MI 48109, USA; 2Gérard Mourou Center for Ultrafast Optical Science, University of Michigan, Ann Arbor, MI 48109, USA; 3Department of Nuclear and Quantum Engineering, Korea Advanced Institute of Science and Technology, Daejeon 34141, Republic of Korea; 4Department of Nuclear Engineering, Seoul National University, Seoul 08826, Republic of Korea; 5Idaho National Laboratory, Idaho Falls, ID 83415, USA

**Keywords:** laser-induced breakdown spectroscopy (LIBS), gamma irradiation, neutron irradiation, advanced reactors, optical absorption

## Abstract

Increased absorption of optical materials arising from exposure to ionizing radiation must be accounted for to accurately analyze laser-induced breakdown spectroscopy (LIBS) data retrieved from high-radiation environments. We evaluate this effect on two examples that mimic the diagnostics placed within novel nuclear reactor designs. The analysis is performed on LIBS data measured with 1% Xe gas in an ambient He environment and 1% Eu in a molten LiCl-KCl matrix, along with the measured optical absorption from the gamma- and neutron-irradiated low-OH fused silica and sapphire glasses. Significant changes in the number of laser shots required to reach a 3σ detection level are observed for the Eu data, increasing by two orders of magnitude after exposure to a 1.7 × 10^17^ n/cm^2^ neutron fluence. For all cases examined, the spectral dependence of absorption results in the introduction of systematic errors. Moreover, if lines from different spectral regions are used to create Boltzmann plots, this attenuation leads to statistically significant changes in the temperatures calculated from the Xe II lines and Eu II lines, lowering them from 8000 ± 610 K to 6900 ± 810 K and from 15,800 ± 400 K to 7200 ± 800 K, respectively, for exposure to the 1.7 × 10^17^ n/cm^2^ fluence. The temperature range required for a 95% confidence interval for the calculated temperature is also broadened. In the case of measuring the Xe spectrum, these effects may be mitigated using only the longer-wavelength spectral region, where radiation attenuation is relatively small, or through analysis using the iterative Saha–Boltzmann method.

## 1. Introduction

There is significant current interest in the development of optical spectroscopy instrumentation for diagnostic applications in advanced reactor systems [1,2,3,4,5]. Laser-induced breakdown spectroscopy (LIBS) has been proposed as a candidate for instrumentation due to its multiple favorable characteristics, such as not requiring sample preparation, sensitivity to a wide range of materials, not requiring radioactive decay for detection, compatibility with analytes of arbitrary phases or compositions, and the ability to take remote measurements [6,7]. In LIBS, a high-power laser pulse, typically in the nanosecond range, is focused onto a sample and produces ionization through the mechanisms of multiphoton ionization and inverse bremsstrahlung [6,8]. In the context of reactor monitoring, researchers have proposed the inclusion of an LIBS module to probe the reactor coolant stream and thus continuously track its composition [9].

The deployment of LIBS instrumentation in nuclear reactors will, in many cases, require that the optical components such as windows, lenses, and fibers be exposed to high doses of ionizing radiation. For successful LIBS implementation, these optics must remain optically transparent to both the excitation source (driving laser) and the plasma emission. It is well-known that optical materials such as glass experience damage at the atomic and molecular levels that results in macroscopic changes to the material properties such as absorption and a change in the refractive index [10,11,12,13]. The altered material absorption changes the measured spectral line intensities when the plasma emission is transmitted through optics. These changes are nonuniform across the optical spectrum; a greater increase in absorption occurs in the shorter (from blue to UV) spectral range. Previous studies [14,15] examined how the altered transmission properties of optical components exposed to gamma radiation result in line attenuation of spectra relevant to nuclear fuel debris analysis.

Here, we present an analysis framework to predict the effects of irradiation on the analysis of the measured LIBS spectra. The calculations use the absorption data from recent irradiation studies [13,16] and two LIBS measurements designed to mimic the conditions that may be observed in advanced reactor systems: (1) a measurement of 1% Xe in an ambient He environment relevant to gas-cooled fast reactor fuel cladding failure monitoring [2] and (2) monitoring 1% Eu in LiCl-KCl, relevant to contamination in coolant streams and in pyroprocessing for molten salt reactors [5,17]. Due to the relatively small increases in absorption at the relevant Xe emission wavelengths, the detectability of individual spectral lines was found to be largely unaffected by 10-Mrad gamma irradiation for both the fused silica glass and sapphire. However, the more pronounced change of transmission induced by exposure to combined neutron and gamma radiation results in a significant increase in the number of laser shots required for a statistically significant line intensity measurement. Similarly, the strong attenuation predicted in the region where Eu emissions occur increases the number of laser shots required to achieve statistical significance for all irradiation conditions examined, with the mixed radiation fluence increasing the required number of laser shots by two orders of magnitude. The spectrally dependent nature of the transmission changes is also found to lead to significant errors in measuring both the Xe/He and Eu/Cl line intensity ratios and the extracted physical parameters such as the plasma temperature, which can be calculated from the relative line intensities. However, if analysis can focus on the lines within the near-infrared region, where the change of attenuation is small, then the predicted effects are negligible. These results improve the understanding of the performance of LIBS systems exposed to large doses of ionizing radiation, which could extend their operational lifetimes and increase the period between instrumental calibrations.

## 2. Materials and Methods

### 2.1. Method and Calculation Details

The reduction in spectral line intensity is determined from the measured spectral absorption, quantified as the transmission *T*: (1)$$T=10^{-D_{A}},$$
where $D_{A}$ is the absorbance in the units of optical density (OD) reported in [13,16]. The measured spectrally dependent absorption was interpolated and multiplied by the measured spectrum. Linear interpolation was used because of the relatively slow point-to-point variation in the measured spectrally dependent transmission. The transmission spectra for the materials studied as a function of the radiation dose can be seen in Figure 1. The effects studied in this work were separated into two categories: those impacted by a single spectral line and those impacted simultaneously by multiple spectral lines.

#### 2.1.1. Single-Line Effects

Single-line detectability is characterized based on the signal-to-noise ratio (SNR), which is defined as
(2)$$SNR\equiv I_{0}/\sigma_{B}.$$

Here, $I_{0}$ is the intensity of the spectral line, and $\sigma_{B}$ is noise, corresponding to the standard deviation of the background. The noise is calculated based on the spread observed in the region nearest to the spectral line of interest devoid of spectral features over a wavelength range equal to the width of the line. For a set of measurements performed with *N* laser shots, the SNR scales as follows: (3)$$SNR=a\;N^{1/2},$$
where *a* represents a fitting constant. Using this relation, the number of laser shots required to achieve the desired SNR can be estimated. For this study, the detectability limit is defined based on the 3σ criterion such that lines are considered observable if their peak intensity is at least three times greater than the standard deviation of the background.

Both here and in all subsequent calculations, the peak line intensity is determined by fitting a spectral line to a Voigt profile distribution [18,19,20], which is given by
(4)$$\frac{I(\lambda )}{I_{0}}=\frac{2}{\pi }\frac{\lambda_{0}}{w_{l}}I_{\lambda }\int_{-\infty }^{\infty }\frac{\exp\left[-\frac{2.772\lambda_{0}^{2}}{w_{g}^{2}}{\left(\frac{\nu }{c}\right)}^{2}\right]d\left(\frac{\nu }{c}\right)}{1+\frac{4}{w_{l}^{2}}{\left[\left(\lambda -\lambda_{0}\right)-\lambda_{0}\left(\frac{\nu }{c}\right)\right]}^{2}}+y_{0}.$$

Here, $\lambda_{0}$ is the wavelength at the line center, $w_{L}$ is the Lorentzian half-width at half-maximum (HWHM), $w_{G}$ is the Gaussian HWHM, ν is the frequency, *c* is the speed of light, and $y_{0}$ is the vertical offset. The background is defined as a nearly constant region of the spectrum near the peak that covers a span equal to the full width at the base of the fitted peak, as determined by Equation (Equation 4). Once the number of shots required to reach the 3σ detection level is known, the time required is calculated by dividing the number of shots by the laser repetition rate. These calculations are performed with the original experimental data collected with a set-up that was not exposed to ionizing radiation and with the data adjusted for the radiation-induced attenuation.

#### 2.1.2. Multi-Line Effects

Optical emission line ratios are commonly used to determine the relative component concentrations in a given elemental matrix [21]. The scaling of the Xe-to-He and Eu-to-Li line intensity ratios was examined as a function of the relative attenuation of the spectral lines used. Parameters such as the plasma temperature, which can be used for normalization, are commonly calculated using the relative line intensities of multiple spectral lines via the Boltzmann plot method [22,23,24]. We compare the standard Boltzmann plot method and the Saha–Boltzmann plot method [25,26]: (5)$$\ln{\left(\frac{I_{0}\lambda }{g_{j}A_{ij}}\right)}^{*}=-\frac{1}{k_{B}T_{p}}E_{j}^{*}+\ln\left(\frac{N_{0}hc}{Z(T_{p})}\right).$$

In this expression, $g_{j}$ is the upper-level degeneracy, $A_{ij}$ is the transition strength, $k_{B}$ is the Boltzmann constant, $T_{p}$ is the plasma temperature, $E_{J}$ is the upper-level energy, $N_{0}$ is the species number density, *h* is Planck’s constant, *c* is the speed of light, and Z(T) is the partition function. Asterisks denote quantities that must be adjusted for ionic transitions [25,26]. This method increases the accuracy by including the spectral lines arising from multiple ionization states. The assumption of local thermodynamic equilibrium (LTE) is supported by the McWhirter criterion;
(6)$$n_{e}\gg 10^{19}{\left(\frac{T_{p}}{e}\right)}^{1/2}{\left(\frac{\Delta E}{e}\right)}^{3},$$
as well as the measurement parameters chosen as described in Section 2.2.1 and agreement in the calculation of the plasma density with and without assuming LTE, as described in Section 3. Here, $n_{e}$ is the plasma electron density, *e* is the fundamental charge, and ΔE is the largest difference between adjacent energy levels. This relation describes the threshold at which the electron collision rates surpass the radiative decay rates by a sufficient degree for LTE to be achieved [20,27]. It is assumed that Stark and instrumental broadening are the most significant contributors to spectral line broadening, allowing the plasma density to be calculated as follows: (7)$$n_{e}=n_{e}^{ref}{\left(\frac{\Delta \lambda }{w}\right)}^{1/m},$$
where $n_{e}^{ref}$ is a reference electron density, Δλ is the Lorentzian component of the spectral line full width at half maximum (FWHM), as determined by Equation (Equation 4), *w* is the line-specific Stark broadening parameter reported in the literature [28,29,30], and *m* is a scaling parameter approximately equal to unity for non-hydrogen lines [31]. If LTE or near-LTE conditions exist, then the plasma density value calculated using Stark broadening parameters should be in good agreement with the value calculated using the Saha–Eggert equation (Equation (Equation 8)) such that
(8)$$n_{e}=2\frac{N_{i}}{N_{j}}\frac{Z_{j}\left(T_{p}\right)}{Z_{i}\left(T_{p}\right)}{\left(\frac{m_{e}k_{B}T_{p}}{2\pi \hbar^{2}}\right)}^{3/2}\exp\left(-\frac{E_{\infty }-\Delta E}{k_{B}T}\right).$$

Here, $N_{i}$ is the atomic population of the *i*th quantum state, Z(T) is the partition function of the *i*th quantum state, *ℏ* is the reduced Planck constant, $E_{\infty }$ is the species ionization energy, and ΔE is a plasma correction factor [20]. Similar to the single-line effects, the multi-line effects are studied using the experimental LIBS data, which are then adjusted for radiation-induced attenuation.

### 2.2. Experiment

#### 2.2.1. LIBS Data Collection

A schematic of the experimental set-up used for the Xe LIBS measurement is shown in Figure 2a. The experimental cell contained a certified mixture of 0.994% Xe in He (99.999% purity) at room temperature and a pressure of 1.00 × 10^5^ Pa (1.00 bar). Prior to filling, the cell was evacuated to a pressure of 10^−5^ Pa (10^−7^ mbar) to prevent contamination from the ambient air. A 1064 nm Nd:YAG laser (Surelite, Continuum) was focused into the cell by a 100 mm focal length lens to induce gas breakdown. The laser produced 10 ns, 250 mJ pulses at a repetition rate of 10 Hz. The resultant LIBS signal was collected using a collimator (CC52, Andor) and directed into an echelle spectrograph (Mechelle, Andor) through a 0.4 mm diameter optical fiber bundle. An intensified CCD (iStar T334, Andor) was used to record the spectra. The timing between the CCD and laser was maintained using a digital delay generator (DG645, Stanford Research Systems). A gate delay of 1 µs was used to minimize the contribution of continuum radiation while still maintaining an environment conducive to LTE conditions. Additionally, a gate width of 1 µs was selected, allowing for minimal spectral line variation over the collection time [27]. Each spectrum resulted from the accumulation of 20 laser shots. Spectral calibrations were performed using an Ar lamp (Pen Light, Oriel).

The set-up used for Eu measurements is shown in Figure 2b. Measurements were performed inside a glovebox to maintain an ambient Ar environment. The experimental cell contained a mixture of 1.017% EuCl_3_ in molten LiCl-KCl. The plasma was excited using an Nd:YAG laser operated at 266 nm (Q-smart with 2nd and 4th harmonic module, Quantel) with a pulse duration of 5 ns, pulse energy of 25 mJ, and repetition rate of 10 Hz. The beam was focused using a plano-convex lens with a focal length of 500 mm. Similar to the Xe measurements, the signal produced was collected via optical fiber and transmitted to an echelle spectrometer (Mechelle, Andor) coupled to an intensified CCD (iStar DH334T-18F-03, Andor), with timing regulated by a digital delay generator (DG645, Stanford Research Systems). The spectra were measured with a gate width of 12.8 µs and gate delay of 0.500 µs, and they resulted from the accumulation of 40 laser shots. Wavelength and intensity calibrations were performed with standard Hg and deuterium lamps, respectively. Further details on the measurement set-up can be found in [5].

#### 2.2.2. Absorption Measurements

The optical absorption was measured by irradiating the Infrasil-302 fused-silica glass and sapphire (Heraeus) in the dry tubes of the ^60^Co irradiator located at the Nuclear Reactor Laboratory at the Ohio State University and in the water pools of the research reactor at the Radiation Science and Engineering Center of Pennsylvania State University. The glasses were selected due to their common use as LIBS optics. For the gamma irradiation, the Infrasil-302 glass received a total dose of 10 Mrad, while the sapphire received a total dose of 3.6 Mrad. For the neutron irradiation, samples were exposed to combined neutron and gamma radiation with fluences of 3.4 × 10^16^ neutrons/cm^2^ (∼42 Mrad) and 1.7 × 10^17^ neutrons/cm^2^ (∼211 Mrad), respectively. Absorption was measured using a broadband light source (DH-2000-BAL, Ocean Insight) and a UV-NIR spectrometer (HR-4000 CG-UV-NIR, Ocean Insight). Further details on this measurement can be found in [13,16].

## 3. Results

Figure 3 shows the calculated effect on the Xe spectrum for light traversing a 1.2 cm thick window after receiving the maximum dosage for each irradiation method, while Figure 4 shows the same effect for the Eu measurement. From here on, “gamma irradiation” denotes the maximum dose received by the sample during gamma-only irradiation, while “neutron irradiation” denotes the highest fluence to which a sample was exposed when combined gamma-neutron irradiation was used. While all cases demonstrated more significant effects at shorter wavelengths, the exposure to high neutron flux was particularly detrimental to the detection of spectral lines in the important region of 250–500 nm, where the majority of Eu and Xe II emissions are found. It can be noted, however, that little effect was observed in the near-infrared spectral region, where Xe I spectral emissions are located. While significant attenuation is unavoidable for Eu analysis, this suggests that it may be beneficial to base the Xe analysis primarily on the Xe I lines, since the attenuation is nearly constant across the spectral range where those lines are located, and over 75% of the line intensity is maintained. In contrast, the results suggest that the Eu II lines located below 300 nm are the least suitable ones for analysis in these conditions, as only 10–20% of light is transmitted after neutron irradiation. The analysis described below was based on the attenuation as predicted by Figure 3 and Figure 4.

Table 1 shows the changes to the single-line detectability as a function of the received radiation dose for select Xe and Eu transitions, as calculated by Equation (Equation 3). For the gamma irradiations, the transmission of all lines over 400 nm remained above 75% for both tested materials, resulting in negligible changes to the number of laser shots required to meet the 3σ detection criterion for lines within this region. However, more significant attenuation was observed for lines located within the UV portion of the spectrum, particularly for Infrasil-302. In contrast, the line attenuation resulting from neutron damage led to up to a ∼100× increase in the minimum number of laser shots to meet the 3σ detection criterion for UV spectral lines and up to a ∼10× increase for visible light spectral lines. Due to the high repetition rate available from modern laser systems, the required measurement time in these cases would remain tolerable. For example, for the laser used in this study, which operated at a modest repetition rate of 10 Hz, all measurement times would remain in the order of minutes.

Figure 5 and Table 2 compare the observed line intensity ratios for the Xe I 828.0-nm emission and Xe II 484.4 nm emission to the He I 587.6 nm emmision as well as the Eu I 281.4-nm emission to the Li I 670.8-nm emission change for different window materials and radiation doses. Due to the increased attenuation at shorter wavelengths, the radiation-induced attenuation reduced the line intensity ratio for the short-wavelength Xe and Eu emissions and increased the line ratio for long-wavelength Xe emissions. Differences in material response can be attributed to radiation damage at molecular sites unique to the glass composition [13,16].

Boltzmann and Saha–Boltzmann plots were constructed by selecting the spectral lines that were resolvable, could be attributed to a single transition, and had relatively high upper-level energies (Table 3) [32]. Figure 6 displays the calculated Boltzmann and Saha–Boltzmann plots for the Xe I and Xe II lines, while Figure 7 displays the Boltzmann and Saha–Boltzmann plots for both the original Eu data and the radiation-induced, attenuation-corrected data. For both glasses, the presence of distinct transmission features led to highly nonuniform attenuation across the spectrum, altering the temperature as calculated from the linear regression of Equation (Equation 5). The LTE assumption was supported by the calculated plasma densities of (1.54 ± 0.35) × 10^17^ cm^−3^ and (5.01 ± 0.33) × 10^17^ cm^−3^ for Xe and Eu, respectively, which met the McWhirter criterion [20,33]. The Xe value was calculated by averaging the plasma densities calculated individually for the 484.4 nm, 541.9 nm, 603.6 nm, and 605.1 nm lines associated with Xe II using the published Stark data [28,30] and the Lorentzian component of their respective FWHM values. The Eu value was calculated analogously using the 397.2 nm and 420.5 nm lines associated with Eu II using the published Stark data [29]. These values were found to be in good agreement with the density calculated by the Saha–Eggert equation (Equation (Equation 8)). The error bars reflect the 95% confidence interval around the calculated temperature, as determined from the slope produced from linear regression. The changes in calculated temperature and the fit error resulting from attenuation corrections are shown in Table 4. When the Xe I lines alone were used, no significant change in temperature was noted due to the nearly uniform reduction in line intensities. As a result, the line slope from which the temperature was calculated remained nearly constant. In contrast, significant changes could be observed when only Xe II lines were used, since they were located in more strongly attenuated spectral regions and spanned a greater spectral range. This effect became even more pronounced for the Eu II lines. While the observed differences could be mitigated if the ionic line corrections were applied within the Saha–Boltzmann plot for Xe, the irradiation-induced attenuation acted to reduce the goodness of fit for the linear regression, as evidenced by the decrease in the associated $R^{2}$ value and increased fitting error. For Eu, while the temperatures determined once the irradiation effects were included were self-consistent, the temperatures predicted were not only significantly lower than the actual temperature but unrealistically low for a plasma to be sustained.

## 4. Discussion

Absorption data from the gamma and neutron irradiation of fused silica glass and sapphire windows were used to investigate the effect on the quantities derived from the analysis of LIBS spectra, in which the emitted light traveled through one of these materials. For an LIBS measurement of Xe in an ambient He environment, it was found that for spectral lines that experienced the most significant attenuation, the required number of laser shots to detect the line with the same statistical certainty increased by one order of magnitude, while the detectability of lines that were minimally attenuated remained largely unchanged. Even at a modest repetition rate of 10 Hz, the measurement times in the investigated case remained relatively short, being to the order of seconds. However, for an LIBS measurement of Eu in molten LiCl-KCl, the severe attenuation led to a more significant increase in the required measurement time from under a second to the order of minutes. The effect of irradiation may have been more pronounced if the analysis were based on less prominent spectral lines. The nonuniform attenuation across the spectrum was found to have more noticeable effects on calculations that relied on the comparison of line intensity ratios, effectively overestimating the Xe/He and Eu/Li line ratios of shorter-wavelength Xe spectral lines such as the 484.4 nm Xe II and 281.9 nm Eu II lines, respectively, and underestimating the Xe/He line ratio of longer-wavelength Xe lines such as the 828.0 nm Xe I line. The nonuniform attenuation effect also manifested in the calculation of temperature using Boltzmann plots such that the error for the resultant temperature value increased.

The results of this study suggest that in the case of spectral lines of interest to helium-cooled fast reactors, the analytical errors introduced by the radiation-induced attenuation can be made small or can be mitigated through additional calibrations. However, it must be noted that the Xe concentration used is significantly greater than what would be expected within a reactor coolant stream during standard operation. Similarly, the concentration of Eu used in this study greatly exceeds what would be present within realistic measurement conditions in molten salt reactors. This likely greatly improved the predicted single-line detectability. To more accurately predict the effects within a reactor environment, the same analysis could be applied to a spectrum resulting from ppm to ppb concentrations of the analyte of interest as well as spectra that include other expected fission fragments to help account for matrix effects [21,37]. In addition, studies suggest that the effects of thermal annealing, which may occur concurrently within the glass, depending on the proximity to the high-temperature reactor core, can repair some of the radiation damage [13,16]. Therefore, corrections must also consider these effects. Other future work may include the analysis of spectra relevant to LIBS in other high-radiation environments, such as for spent fuel cask monitoring [38,39].

## Figures and Tables

**Figure 1 sensors-23-00691-f001:**
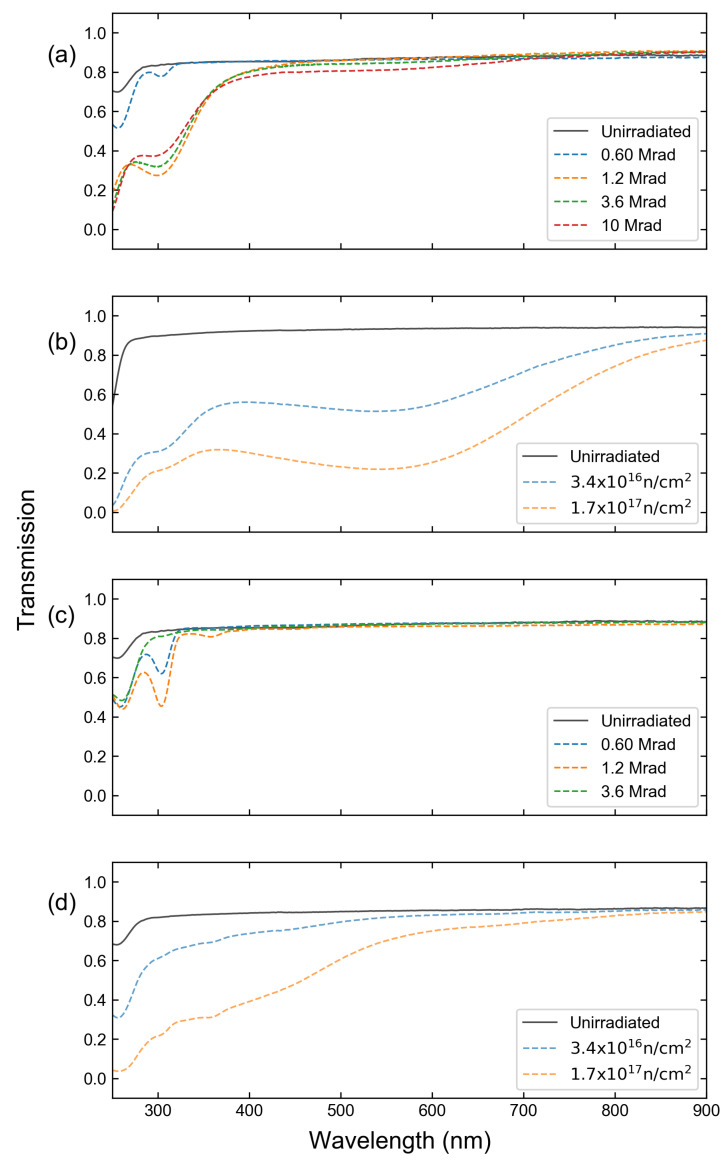
Transmission of (**a**) gamma-irradiated Infrasil-302, (**b**) neutron-irradiated Infrasil-302, (**c**) gamma-irradiated sapphire, and (**d**) neutron-irradiated sapphire windows.

**Figure 2 sensors-23-00691-f002:**
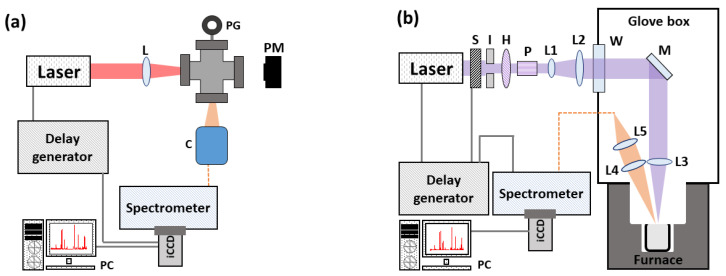
Set-up for (**a**) the Xe in He and (**b**) the Eu in LiCl-KCl LIBS measurements [5]. L = lens, PG = pressure gauge, PM = power meter, C = collimator, S = optical shutter, I = iris diaphragms, H = half-wave plate, P = polarizer, W = window, and M = laser line mirror.

**Figure 3 sensors-23-00691-f003:**
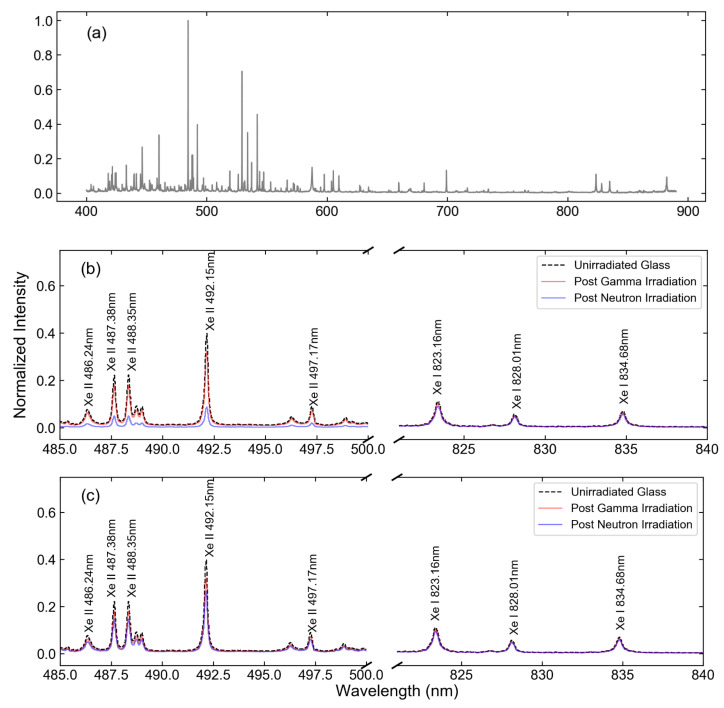
(**a**) Measured 1% Xe in ambient He spectra and spectra adjusted for the changes in absorption in (**b**) Infrasil-302 and (**c**) sapphire windows in the ranges of 485–500 nm and 825–840 nm.

**Figure 4 sensors-23-00691-f004:**
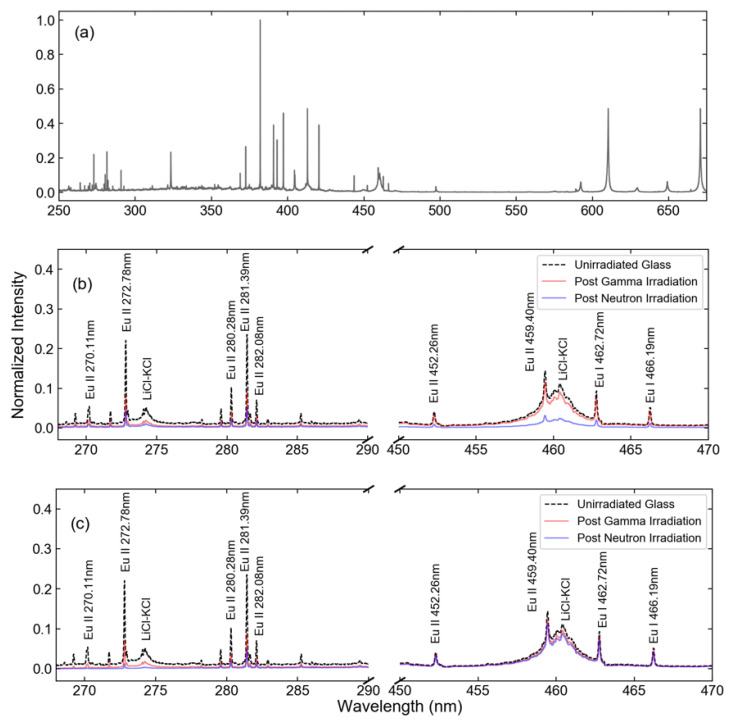
(**a**) Measured 1% Eu in LiCl-KCl spectra and the spectra adjusted for the expected changes in absorption in (**b**) Infrasil-302 and (**c**) sapphire windows in the ranges of 270–290 nm and 450–470 nm. Features resulting from the molten salt background are the result of many closely spaced atomic transitions.

**Figure 5 sensors-23-00691-f005:**
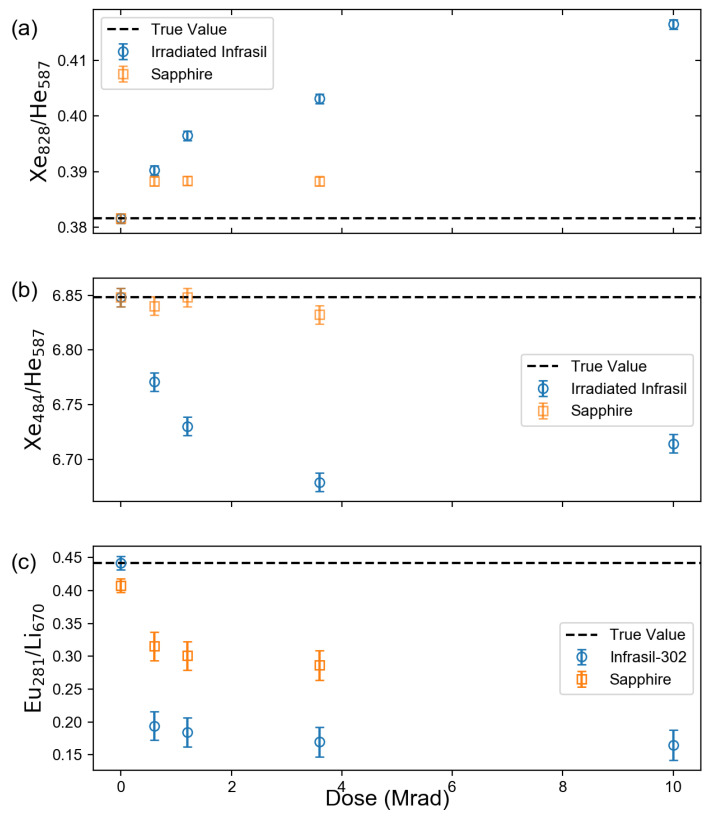
Gamma dose dependence of (**a**) intensity ratio of 828 nm Xe I line and 587 nm He I line; (**b**) intensity ratio of 484 nm Xe II line and 587 nm He I line; and (**c**) intensity ratio of 281 nm Eu I line and 670 nm Li I line.

**Figure 6 sensors-23-00691-f006:**
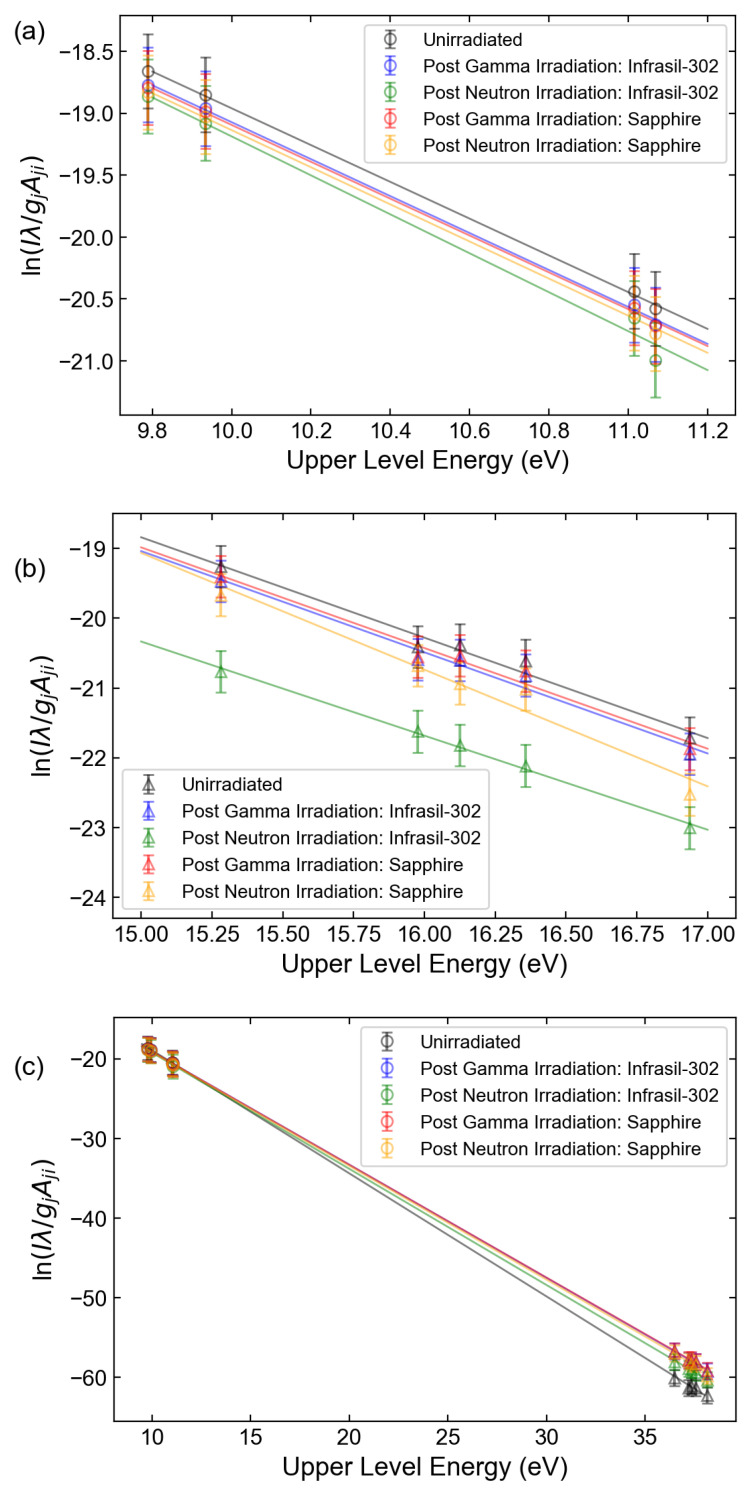
Boltzmann plots using the spectral data both before and after accounting for radiation-induced attenuation for (**a**) Xe I lines only, (**b**) Xe II lines only, and (**c**) both groups combined using the Saha–Boltzmann method.

**Figure 7 sensors-23-00691-f007:**
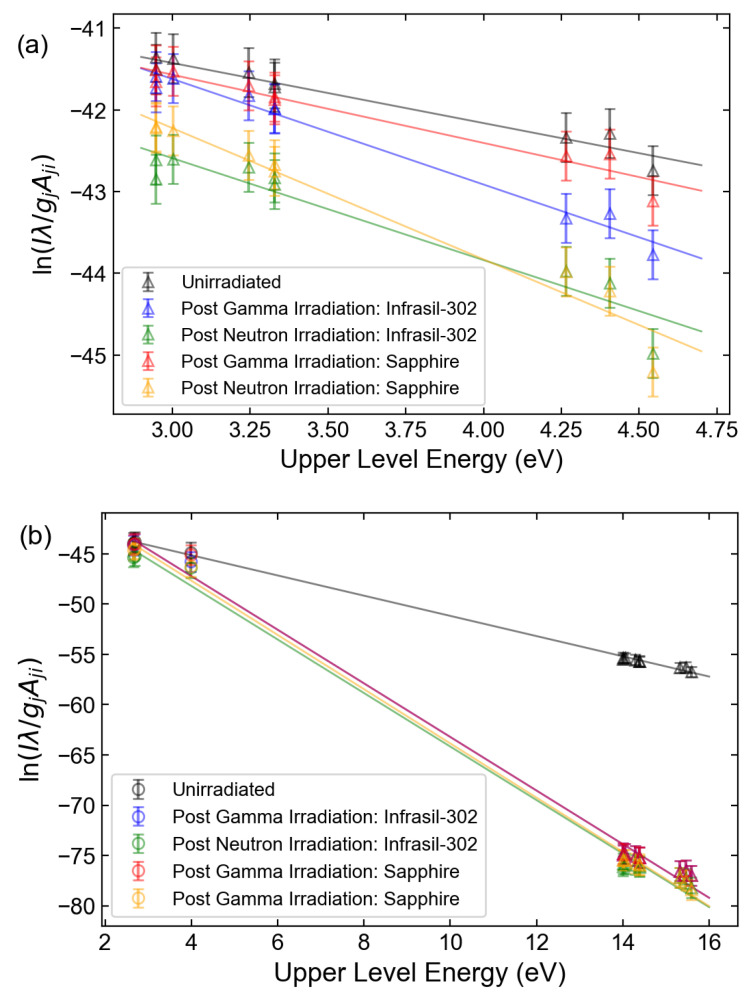
(**a**) Eu II Boltzmann plot and (**b**) Eu Saha–Boltzmann plot.

**Table 1 sensors-23-00691-t001:** The minimum number of laser shots for a 3σ detection level for the examined materials, dependent on the irradiation method and received dose.

Spectral Line	Irradiation Condition	Shots for 3σ
Xe I, 828.0 nm	Unirradiated	1 ± 1
	Infrasil γ Irradiation	2 ± 2
	Infrasil n Irradiation	4 ± 2
	Sapphire γ Irradiation	1 ± 1
	Sapphire n Irradiation	1 ± 1
Xe II, 484.4 nm	Unirradiated	1 ± 1
	Infrasil γ Irradiation	2 ± 2
	Infrasil n Irradiation	17 ± 5
	Sapphire γ Irradiation	1 ± 1
	Sapphire n Irradiation	5 ± 3
Eu I, 462.7 nm	Unirradiated	4 ± 2
	Infrasil γ Irradiation	5 ± 3
	Infrasil n Irradiation	50 ± 7
	Sapphire γ Irradiation	5 ± 3
	Sapphire n Irradiation	13 ± 4
Eu II, 281.4 nm	Unirradiated	6 ± 3
	Infrasil γ Irradiation	36 ± 6
	Infrasil n Irradiation	131 ± 12
	Sapphire γ Irradiation	17 ± 5
	Sapphire n Irradiation	141 ± 12

**Table 2 sensors-23-00691-t002:** Neutron fluence dependence of the line intensity ratios.

Sample	Fluence (n/cm^2^)	I_828_/I_587_	I_484_/I_587_	I_281_/I_670_
Infrasil-302	0 (Unirradiated)	0.382 ± 0.0008	6.74 ± 0.008	0.571 ± 0.01
	3.4 × 10^16^	0.497 ± 0.0008	7.24 ± 0.009	0.248 ± 0.008
	1.7 × 10^17^	1.45 ± 0.0009	9.46 ± 0.009	0.123 ± 0.004
Sapphire	0 (Unirradiated)	0.382 ± 0.0008	6.84 ± 0.008	0.513 ± 0.01
	3.4 × 10^16^	0.509 ± 0.0008	6.51 ± 0.007	0.340 ± 0.009
	1.7 × 10^17^	6.93 ± 0.007	4.57 ± 0.009	0.129 ± 0.004

**Table 3 sensors-23-00691-t003:** Wavelength, transition probability, degeneracy, and upper and lower energy levels of selected Xe and Eu transitions [34,35,36].

Species	Wavelength	Einstein Coeff.	Lower Level		Upper Level	
	(nm)	($10^{7}s^{-1}$)	Degeneracy	Energy (eV)	Degeneracy	Energy (eV)
Xe I	764.202	2.1	1	9.4472	3	11.0691
	828.012	3.69	3	8.4365	1	9.9335
	834.682	4.2	3	9.5697	5	11.0547
	840.919	0.306	5	8.3153	3	9.7893
Xe II	433.052	14	6	14.0737	8	16.9360
	484.433	11	6	11.5390	8	14.0977
	487.650	6.3	6	13.5841	8	16.1259
	526.044	2.2	2	12.9254	4	15.2816
	526.195	8.5	4	14.0009	4	16.3565
	627.082	1.8	4	14.0009	6	15.9775
Eu I	311.143	3.3	8	0.0000	10	3.9836
	462.720	15.6	8	0.0000	8	2.6787
	466.188	15.2	8	0.0000	6	2.6588
Eu II	272.778	6.5	9	0.0000	11	4.5439
	281.393	5.5	9	0.0000	9	4.4048
	290.668	4.1	9	0.0000	7	4.2643
	372.490	4.5	9	0.0000	9	3.3276
	381.967	12.7	9	0.0000	11	3.2450
	397.190	8.9	7	0.2070	9	3.3276
	412.770	6.8	9	0.0000	9	3.0014
	420.505	7.1	9	0.0000	7	2.9476
	452.257	0.99	7	0.2070	7	2.9476

**Table 4 sensors-23-00691-t004:** Comparison of the calculated temperatures for the Xe Boltzmann plots and the Saha–Boltzmann plot.

Method	Irradiation Condition	Temperature (K)	R^2^
Xe I Boltzmann	Unirradiated	7810 ± 390	0.9993
	Infrasil γ Irradiation	7770 ± 390	0.9990
	Infrasil n Irradiation	7370 ± 410	0.9909
	Sapphire γ Irradiation	7800 ± 390	0.9992
	Sapphire n Irradiation	7740 ± 400	0.9987
Xe II Boltzmann	Unirradiated	8000 ± 610	0.9779
	Infrasil γ Irradiation	8000 ± 850	0.9755
	Infrasil n Irradiation	8600 ± 890	0.9960
	Sapphire γ Irradiation	8000 ± 840	0.9759
	Sapphire n Irradiation	6900 ± 880	0.9603
Eu II Boltzmann	Unirradiated	15800 ± 400	0.9551
	Infrasil γ Irradiation	8900 ± 400	0.9813
	Infrasil n Irradiation	9300 ± 600	0.9164
	Sapphire γ Irradiation	13900 ± 400	0.9473
	Sapphire n Irradiation	7200 ± 800	0.9557
Xe Saha-Boltzmann	Unirradiated	7900 ± 350	0.9983
	Infrasil γ Irradiation	8100 ± 460	0.9981
	Infrasil n Irradiation	7660 ± 500	0.9978
	Sapphire γ Irradiation	8000 ± 410	0.9981
	Sapphire n Irradiation	8100 ± 490	0.9981
Eu Saha-Boltzmann	Unirradiated	15200 ± 400	0.9986
	Infrasil γ Irradiation	4400 ± 500	0.9958
	Infrasil n Irradiation	4300 ± 400	0.9947
	Sapphire γ Irradiation	4400 ± 500	0.9926
	Sapphire n Irradiation	4300 ± 400	0.9969

## Data Availability

The data that support the findings of this study are available from the corresponding author upon reasonable request.

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
