# Peer review of "Impact of Glass Irradiation on Laser-Induced Breakdown Spectroscopy Data Analysis"

_sensors, 2023, doi:10.3390/s23020691_

Round 1

Reviewer 1 Report

The authors investigated the effect of glass irradiation on line attenuation in laser-induced breakdown spectroscopy diagnostics. They presented transmission spectra of two materials (Infrasil-302 and sapphire) under gamma and neutron rays. Also, LIBS spectra of Xe and Eu were listed at different irradiation conditions, and some lines were used to calculated the plasma temperatures. This type of research is rare. I think readers in the LIBS field are interested in it. Some comments are as following:

1. Please specify the relationship between Figure 1 and Equautin 1.

2. For Figures 3 and 4, what is the radiation dose of Gamma and Neutron?

3. What are the data acquisition conditions of Figures 5, 6 and 7? Please specify!

4. Whether the spectral lines in Figures 3 and 4 can be calibrated by Figure 1?

5. The change in the line intensity for different irradiations in figure 3c is slight compared with figure 4c, why?

6. For Table 4, whether the temperature can be calibrated to the correct temperature? If only some wrong temperature is provided, I think the comparison is meaningless.

7. The authors also showed the calculation of electron density, such as equation 7. Had the authors evaluated the change in the electron density?

8. Page 14 and line 281, there's an extra “?”.

Author Response

Dear Reviewer,

My co-authors and I are submitting a revised version of our manuscript previously titled ”Impact of Glass Irradiation on Line Attenuation in Laser-Induced Breakdown Spectroscopy Diagnostics.” After some discussion amongst the co-authors, we would like to slightly alter the title to ”Impact of Glass Irradiation on the Analytical Performance of Laser-Induced Breakdown Spectroscopy Diagnostics”, as we feel this more accurately captures the contents of the manuscript. We appreciate the opportunity to re-submit the revised manuscript and for the constructive feedback offered by the reviewers. The comments have been addressed as described below and changes within the manuscript are highlighted.

1. Please specify the relationship between Figure 1 and Equation 1.
The original data as measured by Morgan et al. was expressed in terms of absorption. Using Equation (1), that data is expressed in terms of transmission, which is more intuitive in presenting the effect on the spectral lines shown in Figure 1. Language has been added to the manuscript to clarify this.
2. For Figures 3 and 4, what is the radiation dose of Gamma and Neutron?
The neutron and gamma dose is estimated to be ∼211 Mrad. This value was reported by Morgan et al. in terms of radiation fluence instead of radiation dose. Therefore, Figures 3 and 4 also use the same convention. This has been clarified within the manuscript.
3. What are the data acquisition conditions of Figures 5, 6 and 7? Please specify!
The calculations used to produce Figures 5, 6, and 7 are based on the data displayed in Figures 3 and 4. There is no additional data acquired beyond what is described for these figures. Language has been added to clarify this.
4. Whether the spectral lines in Figures 3 and 4 can be calibrated by Figure 1?
Figures 3 and 4 display the attenuation as predicted by Figure 1. In principle, the transmission curves like those shown in Figure 1 could be used to help calibrate the spectral lines provided the radiation fluence to which the glass was exposed is well-defined. However, as briefly discussed in the conclusion, in realistic application conditions, calibrating the lines from shown transmission curves alone is incomplete; transmission curves that account for the heat to which the glass was exposed would be required, as thermal annealing has been demonstrated to mitigate some radiation-induced attenuation.
5. The change in the line intensity for different irradiations in figure 3c is slight compared with figure 4c, why?
This results from the different spectral regions in which spectral lines were sampled. It can be noted from Figures 1c and 1d that prominent changes in transmission for the sapphire glass only occur below ∼400 nm for the gamma-only irradiation and below ∼500 nm for the combined neutron and gamma
irradiation. Therefore, for the Xe spectral lines displayed in Figure 3c, which occur primarily at longer wavelengths, the attenuation from the glass would be minimal. In contrast, the Eu lines displayed in Figure 4c lie at shorter wavelengths, resulting in significant reductions in spectral line intensity.
6. For Table 4, whether the temperature can be calibrated to the correct temperature? If only some wrong temperature is provided, I think the comparison is meaningless.
Table 4 is meant to provide readers with a quantitative understanding of the magnitude of change to the calculated temperature and linearity when radiation damage effects are introduced. Because there is inherently a relatively large error associated with plasma temperature calculations using the Boltzmann
1 plot method, the difference may be acceptable in some cases, but not others, depending on the objectives for analytical accuracy. The authors are hesitant to comment on how readily the temperature could be calibrated to the correct temperature, as realistically there are additional effects beyond linear absorption changes which must also be considered. These include changes in spectral line attenuation caused by thermal annealing of the glass and changes in spectral line characteristics caused by interference from other constituents of a realistic chemical matrix.
7. The authors also showed the calculation of electron density, such as equation 7. Had the authors evaluated the change in the electron density?
We did not evaluate the change of electron density because the spectral line attenuation occurring from radiation damage does not result in a significant change in the measured line broadening, from which electron density is calculated. However, because the electron density is critical for supporting the local thermodynamic equilibrium assumption and is used in Saha-Boltzmann calculations, Eq. (7), a description of its calculation is included in the manuscript.
8. Page 14 and line 281, there’s an extra “?”.
We thank the reviewer for catching this formatting error. This has been corrected. 

Reviewer 2 Report

1. The abstract is a bit long, please remove some unnecessary sentences

2. In the introduction, please add some similar previous works and show the difference between your current work and these previous works

3. Please add number the equation (Saha-Eggert equation, such that,.....) 

4. Please improve the resolution of Fig.2 

5. Please improve the resolution of Fig.3

6. For the e Boltzmann and Saha-Boltzmann, why you plots some data in figures and added some data in tables?

Author Response

Dear Reviewer 2,

My co-authors and I are submitting a revised version of our manuscript previously titled ”Impact of Glass Irradiation on Line Attenuation in Laser-Induced Breakdown Spectroscopy Diagnostics.” After some discussion amongst the co-authors, we would like to slightly alter the title to ”Impact of Glass Irradiation on the Analytical Performance of Laser-Induced Breakdown Spectroscopy Diagnostics”, as we feel this more accurately captures the contents of the manuscript. We appreciate the opportunity to re-submit the revised manuscript and for the constructive feedback offered by the reviewers. The comments have been addressed as described below and changes within the manuscript are highlighted.

1. The abstract is a bit long, please remove some unnecessary sentences
We thank the reviewer for pointing this out. We revised the abstract length.
2. In the introduction, please add some similar previous works and show the difference between your current work and these previous works
A reference to two previous works, which also investigated the effect of radiation on LIBS for nuclear reactor applications, was added to the introduction.
3. Please add number the equation (Saha-Eggert equation, such that,.....)
This reference has been added.
4. Please improve the resolution of Fig.2
We replaced this image with a higher-resolution version.
5. Please improve the resolution of Fig.3
We also replaced this image with a higher-resolution version.
6. For the e Boltzmann and Saha-Boltzmann, why you plots some data in figures and added some data in tables?
The aim was to demonstrate the changes in the slope and linearity qualitatively by showing the plots while still quantitatively displaying the degradation in the accuracy of the calculated plasma temperature within the tables.

Reviewer 3 Report

The paper deals with the optical spectroscopy instrumentation for diagnostic applications in advanced reactor systems. Laser-induced breakdown spectroscopy (LIBS) is very convenient technique in diagnostics since it is remote, nondestructive, sensitivity, etc. LIBS diagnostics in processes high radiation environments faces with the alternation of material in respect to excitation laser source transmission and absorption that changes the measured spectral line intensities when the plasma emission is transmitted through optics. The current investigation is based on the previous authors’ studies of trace xenon detection in helium environment via LIBS in ref. [2] and changes of optical absorption in quartz glass, fused silica and sapphire optical elements upon neutron and gamma irradiation (refs. [13,14]). The effects are analyzed regarding the impacts on a single spectral line measurements (determination of 3 sigma limit in SNR) and those on simultaneous multiple spectral measurements (intensity ratios of spectral lines used to determine the relative component concentrations). Measurements show significant effects at shorter wavelengths for the exposure to high neutron while little effect is observed in the near-infrared spectral region. One of the conclusions is that the Eu II lines located below 300 nm are the least suitable for analysis, as only 10–20% of light is transmitted after neutron irradiation. Presented are calculated Boltzmann and Saha- Boltzmann plots for the Xe I and Xe II lines and for both the original Eu data and the radiation-induced attenuation- corrected data. For both glasses, the presence of distinct transmission features leads to highly nonuniform attenuation across the spectrum, altering the temperature as calculated from the linear regression. Drawback of this study is that the used concentrations of Xe and Eu exceed those within realistic measurement conditions.

Minor issues:

Reference [13] instead of volume 36 it should be 12.

Reference [14] article number 153945 is missing.

Author Response

Dear Reviewer,

My co-authors and I are submitting a revised version of our manuscript previously titled ”Impact of Glass Irradiation on Line Attenuation in Laser-Induced Breakdown Spectroscopy Diagnostics.” After some discussion amongst the co-authors, we would like to slightly alter the title to ”Impact of Glass Irradiation on the Analytical Performance of Laser-Induced Breakdown Spectroscopy Diagnostics”, as we feel this more accurately captures the contents of the manuscript. We appreciate the opportunity to re-submit the revised manuscript and for the constructive feedback offered by the reviewers. The comments have been addressed as described below and changes within the manuscript are highlighted.

Reference [13] instead of volume 36 it should be 12
Reference [14] article number 153945 is missing.
We thank Reviewer 3 for noting these errors. We made the corresponding corrections.

Round 2

Reviewer 1 Report

The authors carried out active responses and further revised the content of the manuscript. I think the present manuscript can be published in Sensors.